# Moderate-To-Vigorous Intensity Physical Activity and Sedentary Behaviour across Childhood and Adolescence, and Their Combined Relationship with Obesity Risk: A Multi-Trajectory Analysis

**DOI:** 10.3390/ijerph18147421

**Published:** 2021-07-12

**Authors:** Abdulaziz Farooq, Laura Basterfield, Ashley J. Adamson, Mark S. Pearce, Adrienne R. Hughes, Xanne Janssen, Mathew G. Wilson, John J. Reilly

**Affiliations:** 1Physical Activity for Health Group, School of Psychological Sciences & Health, University of Strathclyde, Glasgow G1 1QE, UK; adrienne.hughes@strath.ac.uk (A.R.H.); xanne.janssen@strath.ac.uk (X.J.); john.j.reilly@strath.ac.uk (J.J.R.); 2Aspetar Orthopaedic and Sports Medicine Hospital, Doha 29992, Qatar; 3Population Health Sciences Institute, Newcastle University, Newcastle NE2 4AX, UK; laura.basterfield@newcastle.ac.uk (L.B.); ashley.adamson@newcastle.ac.uk (A.J.A.); mark.pearce@newcastle.ac.uk (M.S.P.); 4Human Nutrition Research Centre, Newcastle University, Newcastle NE4 5PL, UK; 5Institute for Sport Exercise and Health (ISEH), University College London, London W1T 7HA, UK; Mathew.Wilson@hcahealthcare.co.uk

**Keywords:** childhood, adolescent, moderate-vigorous intensity physical activity, sedentary behaviour, obesity, fat mass, group-based trajectories

## Abstract

The combined role of objectively assessed moderate-vigorous intensity physical activity (MVPA) and sedentary behaviour (SB) is unclear in obesity prevention. This study aimed to identify latent groups for MVPA and SB trajectories from childhood to adolescence and examine their relationship with obesity risk at adolescence. From the Gateshead Millennium Study, accelerometer-based trajectories of time spent in MVPA and SB at ages 7, 9, 12, and 15 were derived as assigned as the predictor variable. Fat mass index (FMI), using bioelectrical impedance at age 15, was the outcome variable. From 672 children recruited, we identified three distinct multiple trajectory groups for time spent in MVPA and SB. The group with majority membership (54% of the cohort) had high MVPA and low SB at childhood, but MVPA declined and SB increased by age 15. One third of the cohort (31%) belonged to the trajectory with low MVPA and high time spent sedentary throughout. The third trajectory group (15% of the cohort) that had relatively high MVPA and relatively low SB throughout had lower FMI (−1.7, 95% CI (−3.4 to −1.0) kg/m^2^, *p* = 0.034) at age 15 compared to the inactive throughout group. High MVPA and low SB trajectories when combined are protective against obesity.

## 1. Introduction

Recent data from WHO indicates that global obesity and overweight prevalence has tripled during the last four decades in adults and in children has increased from 4% to 18% [1]. Increased sedentariness and lack of moderate-to-vigorous-intensity physical activity (MVPA) [2], due in part to increasing urbanization worldwide, is one of the key factors that explains rising global obesity prevalence [3]. The amount of energy consumed per minute when lying down at rest is 1 MET. Moderate intensity physical activity such as brisk walking or cycling requires 3–6 METs, and vigorous intensity physical activity such as jogging, aerobic dance, or bicycling uphill produces increases in breathing and heart rate and requires more than 6 METs [4]. Determining the age-related trajectories of MVPA in populations is needed in order to better understand obesity aetiology in children and adolescents, and to inform prevention strategies [5].

A trajectory describes the magnitude and direction of an indicator variable over a course of time or age. Standard statistical approaches for the analysis of longitudinal data such as hierarchical and latent curve modelling can account for individual variability but cannot examine the qualitative dimension to study the possibility of existence of meaningful subgroups within a population that follow distinct developmental trajectories. Group-based trajectory modelling has been available since 1999 [6] but has recently gained attention in the analysis of physical activity trajectories [2,7,8,9,10,11,12]. This is a statistical method for the analysis of longitudinal data to analyse developmental trajectories of a specific measured variable [13]. In contrast to standard methods, group-based trajectory analysis inspects individuals with similar trajectories and categorises them to user-defined finite meaningful subgroups [14].

Recent studies have employed group-based trajectory analysis [2,8,9,10,11,15] to longitudinal physical activity data of children and adolescents, and these have identified distinct trajectories of MVPA over time. These have been informative in describing different patterns of change in MVPA over time, but changes in MVPA do not occur in isolation. Other movement behaviours of relevance to the aetiology of obesity, such as sedentary behaviour, are also simultaneously changing over time [16]. There is increasing recognition of the potential value of considering multiple movement behaviours simultaneously in order to better understand the aetiology of obesity [5] To date, studies of trajectories of objectively-measured movement behaviours over time seldom focused on sedentary behaviour. The recent WHO 2020 guidelines [17] on physical activity have emphasized the need for longitudinal studies with objective measures of sedentary behaviour. The combination of objectively-measured MVPA and sedentary behaviour trajectories have not been described across childhood and adolescence to date, and associations between combined trajectories and adiposity have not been investigated.

An extension of univariate group-based trajectory modelling, multi-trajectory modelling can identify latent meaningful groups of participants that follow similar trajectories across more than one indicator or behaviour of interest [18]. This will take full advantage of the available data in situations where often there is a strong interrelationship or collinearity between behaviours. So rather than analysing such data in sequence, multi-trajectory modelling can address it jointly. Since obesity can been linked with both changes in accelerometer-measured MVPA and accelerometer-measured sedentary behaviour, the application of group-based trajectory analysis can be extended from a simple single group-based trajectory analysis to group-based multi-trajectory modelling that takes into account trajectories across more than one indicator. Therefore, this study aimed to identify multi-trajectory latent groups for time spent in MVPA and sedentary behaviour jointly across childhood and adolescence, and to test for associations between combined movement behaviour trajectories (changes in objectively measured time spent in MVPA and time spent sedentary) and adiposity outcomes during adolescence.

## 2. Materials and Methods

### 2.1. Study Design

The data from this study is from the Gateshead Millennium Study (GMS) that employed a prospective longitudinal cohort study design with 8 years of follow up from childhood and adolescence.

### 2.2. Participants and Settings

The participants of this study were from Gateshead, North East England. The cohort has been described in detail elsewhere [19,20,21], but in brief, the participants were 7–8 years old (hereafter referred to as age 7) when baseline habitual physical activity was assessed objectively during the period October 2006 to December 2007. The objective assessment of physical activity was repeated when the cohort was 9–10 years of age (hereafter referred as age 9) from October 2008 to September 2009), 12–13 years of age (hereafter referred as age 12) from October 2011 to December 2012, and finally at 15–16 years of age (hereafter referred as age 15) during the period September 2014 to September 2015. The sample was broadly socio-economically representative of England. Obesity prevalence and trends were similar to those of the rest of England at the time, with relatively high and slowly increasing prevalence of obesity overall, and obesity risk higher in children and adolescents from families of low socio-economic status than from high socio-economic status [19,20].

### 2.3. Physical Activity Assessment

Physical activity was assessed at all four time points using ActiGraph GT1M accelerometers (Actigraph LLC, Pensacola, FL, USA). The method has been previously [22,23] described in detail, but in summary, participants wore the accelerometer on an elastic belt above the right hip during waking hours for 7 consecutive days. They were permitted to remove them only for sleep or water-based activities such as showering, bathing, and swimming. The device was set to summarize activity in 15 s (epochs) sampling intervals. The physical activity data of the participant was considered only if at least 3 days (2 weekdays and 1 weekend) of valid data were obtained. A day with less than 6 h of accelerometery record was excluded from overall analysis.

The intensity of physical activity was determined using acceptable cut-offs used to define MVPA for children of this age (574 counts per 15 s) [24]. Duration in sedentary time was computed when the accelerometer reading was ≤100 cpm, and >10 min of continuous 0 s was considered as non-wear time. As described previously [11], measures at the first three waves of data collection were measured at the same time of the year (same season) so there was no need for any seasonal adjustment, but due to variation in time of measurement at wave 4, seasonal adjustment was performed with accelerometery measures at age 15.

### 2.4. Anthropometric Proxies for Body Fatness and Body Fatness Measures

Height was measured without shoes using a portable Leicester height measure (Seca, Birmingham, UK) to the nearest 0.1 cm. Weight was measured with light indoor clothing to the nearest 0.1 kg with a Tanita TBF300MA (Chasmors, London, UK). Body mass index (BMI) was derived, and BMI z-scores were calculated using the LMS method from the UK90 reference population [25]. Fat mass (kg and percentage) was calculated using the impedance data from the Tanita TBF300MA, as previously described [20]. Outcome measures for adiposity were used were BMI z-scores and fat mass index (FMI, fat mass/height^2^), as described previously [20].

### 2.5. Study Size

The sample size of the study was based on the numbers available when physical activity and anthropometry measures were first made at age 7 in the cohort (n = 602). Despite the natural attrition over the 8-year period, the available sample can be considered adequate and similar to previous longitudinal studies over such an extended period [8,9,26].

### 2.6. Statistical Methods

Prior to data analysis, all continuous variables were checked for extreme values and outliers. Assumptions for normal distribution were tested using histograms and Shapiro–Wilk tests. No data transformation or data cleaning was required for subsequent analysis. In order to summarize age-related changes in anthropometry and physical activity, linear mixed models were performed using an unstructured covariance structure. Group-based trajectory analysis was performed to identify distinct trajectories or groups of individuals that follow similar trajectories of age-related changes in MVPA and sedentary behaviour separately. In line with the objectives of the study, we also performed a group-based multiple trajectory analysis [14] to identify patterns of trajectories of two dependent continuous variables: time spent in MVPA min/day and time spent in sedentary behaviour min/day jointly. The group-based trajectory model uses the maximum likelihood estimation technique by examining the likelihood or probability of a subject’s repeated observations to belong to a latent group. Hence it can provide the individual’s probability of group membership and probability of data given a group membership.

Group-based multi-trajectory analysis estimates trajectories for two or more than two variables of interest and connects them via conditional probabilities (i.e., in our case, the probability of a specific trajectory for sedentary behaviour given that the participant is following a specified trajectory for MVPA). Several models were constructed starting with 4 groups and then with 3, 2, and 1 group(s) with combinations of linear, quadratic trajectories for MVPA and sedentary behaviour over time (age 7 to 15 years). Only one model that had the smallest value of Bayesian Information Criteria (BIC) was regarded as a best fit model [27]. The models that provided trajectories less than 2% for a trajectory group were not considered, and using a maximum probability assignment rule, each participant was classified under a trajectory for which the membership probability was the highest. Model adequacy was further tested by examining the average posterior probability of each group, which is recommended to be above 0.70 [13]. In addition, the odds of correct classification were determined for each trajectory group. To determine if identified trajectories were associated with body composition at age 15, a one-way analysis of variance (ANOVA) was performed with identified latent groups on the outcomes (BMI z-score, FMI at age 15). Post-hoc analysis with Bonferroni correction was used for multiple comparisons in the event of significant association. A chi-square test was performed to compare the prevalence of obesity at age 15 (BMI z-score ≥ 2.0) across identified latent groups.

## 3. Results

The number of participants at ages 7, 9, 12, and 15 were 602, 585, 525, and 361, respectively; 545 participants provided physical activity data at two time points and 217 at all waves. There were no differences in baseline BMI z-scores and physical activity levels of the participants who were lost to follow up compared to participants who provided data more than once [26].

### 3.1. Characteristics of Study Participants

Table 1 gives the age, anthropometry, and body composition of the study participants at each wave. BMI and FMI were significantly higher at each wave compared to the previous wave. The prevalence of obesity defined by z-score ≥ 2 was 14% at age 15. Mean time spent in MVPA was always below 60 min/day and declined with age from baseline. Conversely, the average time spent in sedentary behaviour per day increased with age from baseline, almost doubling at age 15 compared to age 7. At age 15, FMI was negatively correlated with MVPA (r = −0.235, *p* < 0.001) but not correlated with sedentary behaviour (r = 0.042, *p* = 0.284).

### 3.2. Group-Based Trajectories of Time Spent in MVPA and Sedentary Behaviour Considered Separately

Group-based trajectory analysis of MVPA only revealed four distinct groups (Figure 1). Group 1 had the lowest MVPA at baseline, and it continued to decrease with age (7% of sample). Group 2 had higher MVPA at baseline, but MVPA dropped by age 9 (61% of sample). Group 3 had relatively high MVPA at ages 7 and 9, but MVPA declined from ages 12 onwards (26% of sample). Group 4 had high MVPA at all time points, with the mean above 60 min/day at each time point (6% of the sample). The posterior probabilities for each group were >0.80 and odds of correct classifications (OCC) ≥ 3.

The group-based trajectory analysis for time spent in sedentary behaviour among children found only two distinct groups (Figure 2). Group 1 (45%) was less sedentary, but time spent sedentary gradually increased to age 15. Group 2 consisted of 55% of the children who were more sedentary, and their average time spent in sedentary activities also increased, up to 10 h per day by age 15. The posterior probabilities for each group were >0.81 and OCC ≥ 4.2.

### 3.3. Multiple Trajectory of Time Spent in MVPA and Sedentary Behaviour Simultaneously

The multiple trajectory analysis for time spent in MVPA and sedentary behaviour simultaneously revealed three distinct groups (Figure 3). Group 1 was the most prevalent with 54% of the study sample. They had relatively high MVPA during childhood, but time in MVPA declined and time spent in sedentary behaviour increased with age. Group 2, with 31% of the sample, was characterised by lower MVPA throughout the 8 years (average below 60 min per day) and relatively high daily sedentary time across the 8 years (ranging from 400 to 600 min per day). The third and final group was made up of a small percentage of children (15%) with relatively high MVPA at baseline, but who maintained their levels of MVPA (in contrast to group 1) and who had relatively low sedentary time per day across the 8 years. The posterior probabilities for each group were >0.80 and odds of OCC ≥ 3.0.

### 3.4. Association of Sex with Individual and Multiple Group Trajectories

Girls were more likely to belong to the lowest MVPA throughout the trajectory (80%) compared to 20% for boys, *p* < 0.001 (Table 2). In the high sedentary and increasing trajectory group identified, 56.1% were girls. Similarly, the most active group trajectory for joint trajectories of MVPA and sedentary behaviour had mostly boys (76.4%) compared to 23.6% for girls (*p* < 0.001).

### 3.5. Associations between Individual Group Trajectories and Multiple Group Trajectories and Adiposity at Age 15 Years

FMI at age 15 was significantly associated with individual MVPA trajectory groups. The most active group (Group 4) had lower FMI at age 15 years compared to Group 1 (*p* = 0.012) and Group 3 (*p* = 0.022). Being in the two individual trajectory groups of sedentary time was not associated with FMI at age 15 years (Table 2).

In the multiple group trajectory identified, FMI was significantly higher at age 15 years among the inactive group (low MVPA and high sedentary time throughout) compared to the active group, which had a relatively high MVPA and relatively low sedentary time throughout the 8 years. The observed difference in FMI between inactive vs. active groups was (−1.7, 95% CI (−3.4 to −1.0) kg/m^2^, *p* = 0.034)).

## 4. Discussion

This is the first study to describe the trajectories of objectively assessed co-dependent activity behaviours, time spent in MVPA, and sedentary behaviour, from childhood to adolescence, and the only study to date to test for associations of these combined trajectories with adiposity. The main findings of this study are (a) that there were three distinct trajectories of time spent in MVPA and sedentary behaviour (relatively high but declining MVPA, increasing sedentary time; relatively low and stable MVPA with high and increasing sedentary time; relatively high and stable MVPA with relatively low and stable sedentary time) and (b) that children with high and stable MVPA combined with low and stable sedentary time, had significantly lower fat mass index at late adolescence.

A previous study, from the Iowa cohort that was born around 10 years earlier than the present study GMS cohort, using univariate group-based trajectory analysis (based on changes in MVPA only) found children belonging to an inactive group were more likely to become obese in young adulthood when compared to children who were consistently physically active during childhood and adolescence [9]. The present study extends those findings by showing that both MVPA and sedentary behaviour changes, as co-dependent multi trajectory behaviours are associated with the development of body fatness across childhood and adolescence.

Interestingly, in the present study, although the association between activity and fat mass index was clear, there was no direct relationship of MVPA on body mass index or obesity prevalence at late adolescence either using single group trajectory of MVPA or multi-trajectories of MVPA and sedentary behaviour. Kwon et al. (2015) [9] found that children that decreased MVPA with age were 2.7 times more likely to have ≥80th percentile of percent body fat than those with relatively stable MVPA. The observation that associations with direct measures of body fatness (expressed as percent body fat or fat mass index) can be observed while associations with crude proxies for body composition such as BMI-for-age defined obesity prevalence or BMI-for-age Z-scores emphasizes that, where possible, it is better to have body fatness as an outcome measure rather than proxy of body fatness.

The method of identifying latent groups in the field of physical activity in paediatric populations has always been univariate, that is, looking at only one exposure trajectory of physical activity [7], MVPA [8,9,11,28], questionnaire-based PA [28,29,30], or sports participation [9,10]. The trajectory of sedentary behaviour has rarely been reported [31]. In the present study, we identified four distinct groups for MVPA and three latent groups of sedentary behaviour. In addition to this, we also performed multi-trajectory modelling that has recently been recommended when there is a linkage between two or more distinct behaviours of interest [14]. Since multigroup trajectory analysis of objectively assessed physical activity and sedentary behaviour has not been used to date to study the aetiology of obesity in children and adolescents, there are no directly comparable studies.

Recently two studies using the multi-trajectory analyses of activity behaviours have been published in childhood or adolescence [31,32], but these have been based on self-reported measurements of MVPA and sedentary behaviour, which gives some uncertainty about amounts or intensities of behaviour, and they have been purely descriptive. Although these studies identified latent groups, they did not test whether belonging to these different trajectories influenced any important health outcomes.

Gallant et al. (2020) [32] performed multi-trajectory modelling of behavioural co-development for sleep, self-reported MVPA, and self-reported screen time for Canadian children from age 9 to 11 years to age 17 to 19 years. They identified four latent groups each for boys and girls. The prevalence of a ‘good adherence’ group that met the recommendations for time spent in PA and screen time was 12% and 9% in boys and girls, respectively. This may not be directly comparable to our study that used objectively assessed MVPA and sedentary time rather than self-reported screen time. Nevertheless, prevalence of high MVPA and low sedentary time at age 7 and who remained consistently active at age 15 y in the present study was quite low (14.8%) and comparable to the Canadian study. Taken together with the present study, it seems that only a small minority of children and adolescents have MVPA and sedentary time trajectories that are protective against high body fatness.

A recent study among South African children, which included sedentary time along with movement behaviours such as informal activity, organized sports, walking, and sleep, in order to identify latent trajectories through ages 13 to 17 years, found four distinct groups for boys and two distinct groups for girls. The prevalence of the ‘most adherent group’ that was consistently active with high informal activity and low sedentary behaviour was 7%, around half of the size found in the present study (15%). The differences in prevalence to the present study can probably be attributed largely to objective measurement of the behaviours in the present study, but the studies are similar in that the group with the most favourable combination of physical activity and sedentary behaviour was also the smallest group. Despite marked differences between studies in the time-period when the measures were made, the methodology for measuring movement behaviours, and settings, the findings from the present study, the Iowa Bone Health Study [9,28], and the two very recent studies from South African and North American cohorts [31,32] generally confirm that the proportion of children who are active in childhood and who remain active into middle–late adolescence is alarmingly low, and the combination of adequate time spent in MVPA and low time spent in sedentary behaviour is rare.

Based on the high level of evidence coming from meta-analyses of adult studies, sedentary behaviour is associated with increased odds for obesity (OR 1.3 (95% CI 1.1–1.6) in adulthood [33]. Another meta-analysis of a dose–response relationship showed 1 h/day of sedentary behaviour increases the risk of Type 2 diabetes by 5%, hypertension by 4%, and overweight/obesity by 38% when sedentary behaviour exceeds 3 h/d) [34]. Therefore, interventions should begin early and target all these behavioural and environmental domains. Interventions to reduce sedentary time in children have been fairly successful in short term (<6 months) as well as in long term (≥6 months) studies (−25.9 (95% CI −40.8, −11.0) min and −14.0 (95% CI −19.5, −8.6) min, respectively) [35].

In the present study, the time spent in sedentary behaviour increased with age in all latent groups identified, even among the most active group. Among the studies that looked at sedentary behaviour using objective measures as in this study, using self-reported sedentary behaviour in the South African cohort [31] and self-reported screen time and TV viewing from Canadian [32] and IOWA cohorts [9], all indicate that mean time spent in sedentary behaviour is always increasing with age, and no distinct trajectory was identified in any of these studies that showed reduced sedentary behaviour or screen time with age.

In contrast to increasing trends of sedentary behaviour, time spent in MVPA is generally declining from early–mid childhood. A meta-analysis of the relative change in MVPA per year among children and adolescents showed a decline of −3.4% (95% CI, −5.9 to −0.9) in boys and −5.3% (95% CI, −7.6 to −3.1) in girls, [26]. The latent groups of MVPA identified in our study also shows that MVPA is in decline from an early age. The IOWA cohort recently reported distinct trajectories for MVPA beginning from adolescence to young adulthood. The group characterised by moderately-active participants with decreasing MVPA was the largest (~89% in girls and ~66% in boys), and the second smaller group defined by consistently-active participants with high MVPA was significantly associated with lower FMI at age 15 and age 23. The negative relationship of MVPA trajectory membership with FMI in this study was consistent with our findings that showed a similar negative relationship. Using the multi-trajectory analysis, the average difference in FMI between the ‘active throughout’ group compared to the ‘inactive throughout’ was 1.7 (95% CI 1.0 to 3.7) kg/m^2^. This difference can be considered clinically or biologically important given that an FMI increase of 1 kg/m^2^ is associated with increased risk of ischemic stroke (OR 1.38), heart failure (OR 1.22), and coronary artery disease (OR 1.08), and several other cardiovascular disorders by adulthood [36].

The present study suggests that it is vital to address MVPA to prevent fat gains, and multi-trajectory analysis provides the patterns of sedentary behaviour trajectory in these groups with varying MVPA structures. The joint association of MVPA and sedentary behaviour in our study to fat mass could be as a result of indirect displacement of time spent in active behaviour and will support the argument that reducing sedentary behaviour indirectly reduces fat mass gains.

The main strengths of the present study were the application of a multi-trajectory modelling approach to identify joint trajectories of co-dependent activity patterns, MVPA and sedentary behaviour from early childhood to adolescents, plus the objective measures of the movement behaviours, and the novelty of our tests for associations of these combined trajectories with adiposity outcomes.

The present study also had a number of weaknesses. Unfortunately, the analysis could not be performed separately for boys and girls due to the limited data available at age 15. There were 142 (21.1%) children lost to follow up at age 15, a pattern which is consistent with previous longitudinal studies with follow up greater than 5 years involving children and adolescents [7,9], but loss to follow-up was not differential as noted above. In this analysis, therefore we considered only cases with ≥2 time points, and hence it did not affect the estimates of the identified trajectories, which assumes missingness is random by default [13,27]. However, it must always be acknowledged that identified trajectories in this study cannot necessarily be generalized to other settings internationally or across different time spans; however, based on comparison with other similar studies, it can be highlighted that the most active groups are smaller in size, and the overall absolute amounts of time spent in MVPA and sedentary behaviour and changes over time were similar to other comparable longitudinal studies, as noted above. The relationship for group-based trajectories with outcomes are temporal [14] and must be interpreted with caution. Sex as a factor was strongly associated with both single and multi-trajectory group memberships, with girls more likely to belong to low active trajectories; however, more factors could have been considered, such as socio-economic status, lifestyle behaviours, sports participation, and school commuting. However, due to the overwhelming number of findings from this study, it was decided to analyse the predictors associated with group membership as a separate focused paper.

This study found that MVPA trajectories across childhood and adolescence are strongly associated with adolescent fat mass, reiterating the importance of MVPA in obesity aetiology and prevention [17,37]. The negative correlation of MVPA with FMI is evident from this study. Furthermore, based on the findings from this study, accelerometer-measured sedentary behaviour should also be integrated with MVPA in future interventions targeting obesity. Sedentary behaviour is complex and has multiple facets, including types and settings where it can be accumulated in various ways when the child is at school, home, during commuting, as well as during leisure activities [38]. With the onset of the COVID-19 global pandemic, the demand for addressing sedentary behaviour is now even more urgent, given that activity among school children is compromised [39] and time spent sedentary has increased [40,41].

## 5. Conclusions

The present study suggests that FMI is a better indicator of adiposity in aetiological studies than the BMI-for-age, and that the novel multi-trajectory analysis is a useful way of understanding obesity aetiology by identifying groups with distinct trajectories, but future research is warranted to determine predictors that influence group membership.

## Figures and Tables

**Figure 1 ijerph-18-07421-f001:**
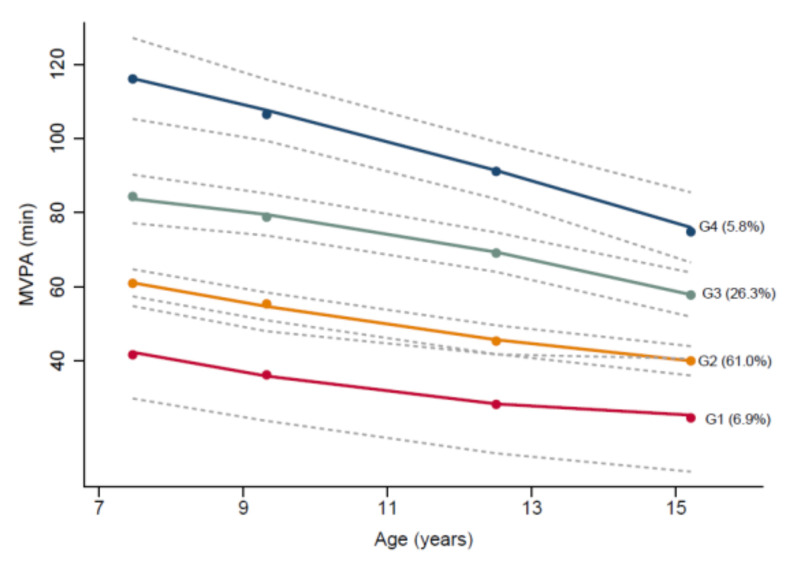
Group based trajectories of time spent in MVPA. Dots indicate average MVPA (minutes/day), and four black lines are trajectories showing estimated mean MVPA (minutes/day) across age for the four groups identified (G1–G4), and grey dashed lines represent the 95% confidence intervals for each trajectory.

**Figure 2 ijerph-18-07421-f002:**
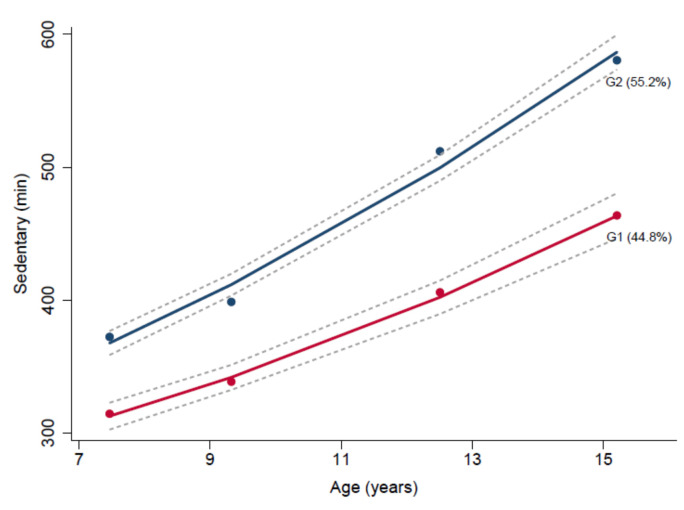
Group based trajectories of time spent in sedentary behaviour. Dots indicate average sedentary behaviour (minutes/day), and two black lines are trajectories showing estimated mean time spent in sedentary behaviour (minutes/day) across age for the two groups identified (G1–G2), and grey dashed lines represent the 95% confidence intervals for each trajectory.

**Figure 3 ijerph-18-07421-f003:**
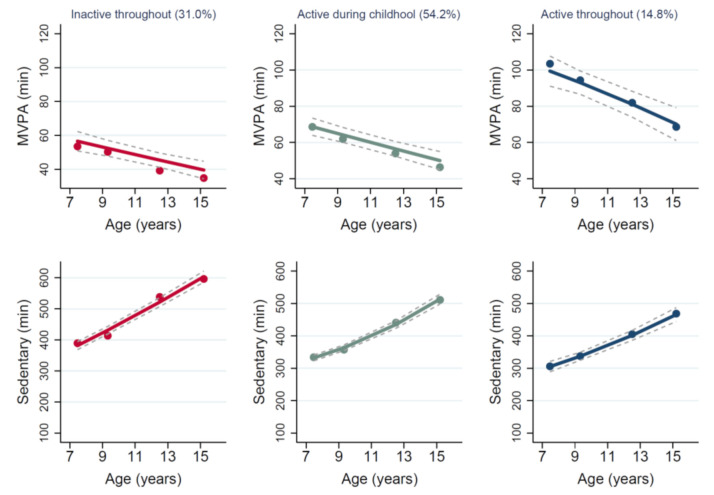
Multiple trajectories of average time spent in moderate-vigorous-intensity physical activity (MVPA) and sedentary behaviour (minutes per day). Dots indicate observed average MVPA and sedentary behaviour (minutes/day), and solid lines are trajectories showing estimated mean time spent in MVPA and sedentary behaviour (minutes/day) across age for the three groups identified. The grey dashed lines represent the 95% confidence intervals for each trajectory.

**Table 1 ijerph-18-07421-t001:** Anthropometry and physical activity (Mean ± SE) of study participants at the four waves by sex.

Variables	Boys				Girls			
	Age 7 n = 279	Age 9 n = 277	Age 12 n = 246	Age 15 n = 173	Age 7 n = 283	Age 9 n = 287	Age 12 n = 260	Age 15 n = 155
**Anthropometry**								
Body mass index z-score *	0.4 ± 0.1	0.6 ± 0.1 ^a^	0.7 ± 0.1 ^a,b,d^	0.6 ± 0.1 ^‡^	0.4 ± 0.1	0.6 ± 0.1 ^a^	0.7 ± 0.1 ^a,b^	0.8 ± 0.1 ^a,b^
Fat mass index	4.1 ± 0.1	4.7 ± 0.1 ^a^	5.2 ± 0.2 ^a,b,‡^	5.1 ± 0.3 ^a,‡^	4.0 ± 0.1	5.0 ± 0.1 ^a^	5.9 ± 0.2 ^a,b^	8.4 ± 0.2 ^a,b,c^
Obesity ** (%)	9.7%	13.4% ^a^	13.8%	13.3%	8.8%	8.4%	12.7% ^a,b^	14.8% ^a,b^
**Accelerometer**								
MVPA (min/day)	75.2 ± 1.4 ^‡,b,c,d^	70.0 ± 1.3 ^‡,c,d^	60.1 ± 1.5 ^‡,d^	51.0 ± 1.5 ^‡^	63.1 ± 1.4 ^b,c,d^	56.3 ± 1.3 ^c,d^	46.7 ± 1.4 ^d^	40.6 ± 1.4
Sedentary behaviour (min/day)	338.3 ± 4.3 ^‡^	364.5 ± 3.9 ^a,‡^	447.5 ± 6.0 ^a,b,‡^	518.7 ± 6.8 ^a,b,c^	352.5 ± 4.3	378.3 ± 3.8 ^a^	480.5 ± 5.7 ^a,b^	536.1 ± 6.4 ^a,b,c^

n: no of children with valid anthropometry data, MVPA: moderate-vigorous intensity physical activity, * relative to UK90 reference data [25]. ** defined as BMI z-score ≥2, results based on linear mixed models after adjusting for multiple comparison, ^a^: significantly higher than age 7, ^b^: significantly higher than age 9, ^c^: significantly higher than age 12, ^d^: significantly higher than age 15 all at *p* < 0.05, ‡: significantly different than girls at the same age.

**Table 2 ijerph-18-07421-t002:** Outcomes at age 15 in association with distinct trajectory groups.

	Univariate Group-Based Trajectories for Moderate–Vigorous-Intensity Physical Activity (MVPA)				
	Group 1 The Lowest MVPA throughout (n = 15)	Group 2 High MVPA but Declining (n = 94)	Group 3 Low MVPA and Declining (n = 221)	Group 4 High MVPA (n = 26)	*p*-Value
Boys	3 (20.0)	56 (59.6)	90 (40.7)	24 (92.3)	<0.001
Girls	12 (80.0)	38 (40.4)	131 (59.3)	2 (7.7)	
Fat mass (kg)	22.5 ± 9.0 ^a^	17.4 ± 12.0	18.7 ± 12.0 ^a^	12.2 ± 6.3	0.021
Fat mass index (FMI)	8.5 ± 3.5 ^a^	6.2 ± 4.0	6.8 ± 4.3 ^a^	4.3 ± 2.3	0.006
BMI Z-score for age	0.9 ± 1.3	0.6 ± 1.1	0.6 ± 1.3	0.3 ± 1.2	0.337
Obesity (BMI SD-score ≥ 2)	26.7%	13.8%	14.0%	7.7%	0.415
	**Univariate Group-Based Trajectories for Sedentary Behaviour**				
	**Group 1** **Low Sedentary Increasing** **(n = 151)**	**Group 2** **High Sedentary and Increasing** **(n = 205)**			***p*-Value**
Boys	83 (55.0)	90 (43.9)			0.038
Girls	68 (45.0)	115 (56.1)			
Fat mass (kg)	17.4 ± 11.7	18.6 ± 11.7			0.515
Fat mass index (FMI)	6.3 ± 4.2	6.6 ± 4.0			0.477
BMI Z-score for age	0.7 ± 1.3	0.6 ± 1.2			0.336
Obesity (BMI SD-score ≥ 2)	15.9%	12.7%			0.389
	**Multi-Trajectory Groups (MVPA and Sedentary Behaviour Combined)**				
	**Group 1** **Inactive throughout** **(n = 110)**	**Group 2** **Active during Childhood** **(n = 191)**	**Group 3** **Active throughout** **(n = 55)**		***p*-Value**
Boys	40 (36.4)	91 (47.6)	42 (76.4)		
Girls	70 (63.6)	100 (52.4)	13 (23.6)		
Fat mass (kg)	19.5 ± 11.5	18.2 ± 12.2	14.9 ± 9.7		0.060
Fat mass index (FMI)	7.0 ± 3.9 ^b^	6.6 ± 4.3	5.3 ± 3.4		0.038
BMI Z-score for age	0.6 ± 1.3	0.7 ± 1.2	0.6 ± 1.1		0.770
Obesity (BMI SD-score > 2)	12.7%	15.7%	10.9%		0.594

One way analysis of variance (ANOVA) was used to compare means between groups, and chi-square test was used to compare proportions across groups. ^a^: significantly higher than Group 4; ^b^: significantly higher than Group 3 at *p* < 0.05.

## Data Availability

The datasets used and/or analysed during the current study are available from the corresponding author on request.

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
