# Peer review of "Moderate-To-Vigorous Intensity Physical Activity and Sedentary Behaviour across Childhood and Adolescence, and Their Combined Relationship with Obesity Risk: A Multi-Trajectory Analysis"

_ijerph, 2021, doi:10.3390/ijerph18147421_

Round 1
Reviewer 1 Report
The manuscript proposed by Farooq A. et al aims to identify multi-trajectory latent groups for time spent in MVPA and sedentary behavior over time from childhood to adolescence.
Data are from a well-designed longitudinal cohort study with 8-year follow-up. The manuscript is weel written and the approach is interesting. However I have a major concern about the two trajectories used in the multi-trajectory latent group approach.
Indeed, the group based multi-trajectory modelling used here is based on the probability of a specific trajectory for sedentary behaviour given the participant is following a specified trajectory of MVPA. This approach is really interesting in exploration of patient trajectories; however, this seem to be rather surprising to use two highly collinear variables for this approach. Indeed, MVPA and sedentarity are defined on the basis of data collected from the accelerometer. So the conclusion seem evident: more sedentary groups will be children who had the lowest MVPA trajectory.
Did the authors performed correction for multiple test when multiple groups were compared? This is not specified in method or in the table footnote. And it is not clear where the differences were observed.
Minor:
Figure 2: the numerous associated to the groups should be added in the graph titles. Indeed, in text Group 1 refer to the most prevalent groups which is in the middle of Figure 2.
Author Response
Reviewer 1:
Open Review
(x) I would not like to sign my review report
( ) I would like to sign my review report
English language and style
( ) Extensive editing of English language and style required
( ) Moderate English changes required
( ) English language and style are fine/minor spell check required
(x) I don't feel qualified to judge about the English language and style
|
Yes |
Can be improved |
Must be improved |
Not applicable |
|
|
Does the introduction provide sufficient background and include all relevant references? |
(x) |
( ) |
( ) |
( ) |
|
Is the research design appropriate? |
( ) |
(x) |
( ) |
( ) |
|
Are the methods adequately described? |
( ) |
(x) |
( ) |
( ) |
|
Are the results clearly presented? |
(x) |
( ) |
( ) |
( ) |
|
Are the conclusions supported by the results? |
(x) |
( ) |
( ) |
( ) |
The manuscript proposed by Farooq A. et al aims to identify multi-trajectory latent groups for time spent in MVPA and sedentary behavior over time from childhood to adolescence.
Data are from a well-designed longitudinal cohort study with 8-year follow-up. The manuscript is weel written and the approach is interesting. However I have a major concern about the two trajectories used in the multi-trajectory latent group approach.
Indeed, the group based multi-trajectory modelling used here is based on the probability of a specific trajectory for sedentary behaviour given the participant is following a specified trajectory of MVPA. This approach is really interesting in exploration of patient trajectories; however, this seem to be rather surprising to use two highly collinear variables for this approach. Indeed, MVPA and sedentarity are defined on the basis of data collected from the accelerometer. So the conclusion seem evident: more sedentary groups will be children who had the lowest MVPA trajectory.
There is always negative correlation between MVPA and sedentary behaviour and the strength of this correlation is usually moderate (For example in one study r=-0.45 Gomes et al., 2017). However, unlike regression analysis, group based multi-trajectory analysis is not affected by the strength of the correlation. For example, if the correlation is very high (>-0.9), the no of groups identified for MVPA will be similar to the number of groups identified by Sedentary Behaviour. In our paper, we have conducted single variable MVPA trajectories, single variable MVPA trajectories and joint trajectories of MVPA and sedentary behaviour. The advantage of this form of joint-trajectory modeling is that it highlights heterogeneity in the linkage between trajectories of distinct outcomes that are thought to be related by a common underlying etiological process.
Group-based multi-trajectory analysis is specially designed to address two concurrently occurring behaviours (Nagin et al., 2016). Recently studies have applied the same strategy For example Gallant et al., 2020 had looked at multiple trajectories for (MVPA, screen time and Sleep). And Hanson et al., 2020 had examined multi-trajectory models for (overall movement behaviour defined by the 3 physical activity domains, sedentary behaviour and school-night sleep). So we assure you that the presence of collinearity should not be of any concern. We have now clarified this in the manuscript (line 89).
Hanson, S. K., Munthali, R. J., Micklesfield, L. K., Lobelo, F., Cunningham, S. A., Hartman, T. J., Norris, S. A., & Stein, A. D. (2019). Longitudinal patterns of physical activity, sedentary behavior and sleep in urban South African adolescents, Birth-To-Twenty Plus cohort. BMC Pediatrics, 19(1), 241. https://doi.org/10.1186/s12887-019-1619-z
Gomes, T. N., Hedeker, D., Dos Santos, F. K., Souza, M., Santos, D., Pereira, S., Katzmarzyk, P. T., & Maia, J. (2017). Relationship between Sedentariness and Moderate-to-Vigorous Physical Activity in Youth: A Multivariate Multilevel Study. International journal of environmental research and public health, 14(2), 148. https://doi.org/10.3390/ijerph14020148
Gallant, F., Thibault, V., Hebert, J., Gunnell, K. E., & Bélanger, M. (2020). One size does not fit all: identifying clusters of physical activity, screen time, and sleep behaviour co-development from childhood to adolescence. Int J Behav Nutr Phys Act, 17(1), 58-58. https://doi.org/10.1186/s12966-020-00964-1
Nagin, D. S., Jones, B. L., Passos, V. L., & Tremblay, R. E. (2016). Group-based multi-trajectory modeling. Stat Methods Med Res, 27(7), 2015-2023. https://doi.org/10.1177/0962280216673085
Did the authors performed correction for multiple test when multiple groups were compared? This is not specified in method or in the table footnote. And it is not clear where the differences were observed.
Yes, we have applied Bonferroni correction, when multiple groups were compared. We have now added this in methods and also included them in the results tables.
Minor:
Figure 2: the numerous associated to the groups should be added in the graph titles. Indeed, in text Group 1 refer to the most prevalent groups which is in the middle of Figure 2.
This is a reasonable suggestion. However, kindly note that Group based trajectory analysis provides latent groups and the size of the groups are estimated from the sample population provided with varying numbers at each wave. Therefore, generally it is acceptable not to show numbers in graphs. Previous papers also follow same recommendations by Nagin et al., 2016; Gallant et al., 2020; Gomes et al., 2017; Hanson et al., 2019.
Overall we are grateful for your valuable suggestions and this has improved the quality of the paper.

Reviewer 2 Report
Comments to Authors
Moderate-to-vigorous intensity physical activity and sedentary behaviour across childhood and adolescence, and their combined relationship with obesity risk: a multi-trajectory analysis
General Comment: The authors purpose to identify multi-trajectory latent groups for time spent in MVPA and sedentary behaviour jointly across childhood and adolescence, and to test for associations between combined movement behaviour trajectories (changes in objectively measured time spent in MVPA and time spent sedentary), and adiposity outcomes during adolescence. The study is an elegant and important, and showed that FMI could be a better indicator of adiposity than the BMI-for-age. However, I have some considerations:
Introduction section: please include the concept and reference of physical activity, moderate and vigorous physical activity and stablish the difference between the intensities (moderate and vigorous).
Discussion and conclusion sections: please include some consideration about the influence and correlation of MVPA on the FMI.
Include the name of statistical tests and values descriptions in all figures and tables.
Author Response
Review 2:
Open Review
(x) I would not like to sign my review report
( ) I would like to sign my review report
English language and style
( ) Extensive editing of English language and style required
( ) Moderate English changes required
( ) English language and style are fine/minor spell check required
(x) I don't feel qualified to judge about the English language and style
|
Yes |
Can be improved |
Must be improved |
Not applicable |
|
|
Does the introduction provide sufficient background and include all relevant references? |
(x) |
( ) |
( ) |
( ) |
|
Is the research design appropriate? |
(x) |
( ) |
( ) |
( ) |
|
Are the methods adequately described? |
(x) |
( ) |
( ) |
( ) |
|
Are the results clearly presented? |
(x) |
( ) |
( ) |
( ) |
|
Are the conclusions supported by the results? |
(x) |
( ) |
( ) |
( ) |
Comments and Suggestions for Authors
Comments to Authors
Moderate-to-vigorous intensity physical activity and sedentary behaviour across childhood and adolescence, and their combined relationship with obesity risk: a multi-trajectory analysis
General Comment: The authors purpose to identify multi-trajectory latent groups for time spent in MVPA and sedentary behaviour jointly across childhood and adolescence, and to test for associations between combined movement behaviour trajectories (changes in objectively measured time spent in MVPA and time spent sedentary), and adiposity outcomes during adolescence. The study is an elegant and important, and showed that FMI could be a better indicator of adiposity than the BMI-for-age. However, I have some considerations:
Introduction section: please include the concept and reference of physical activity, moderate and vigorous physical activity and stablish the difference between the intensities (moderate and vigorous).
We have now defined moderate and vigorous intensity physical activity in the paper. (Lines 51-55)
Discussion and conclusion sections: please include some consideration about the influence and correlation of MVPA on the FMI.
At age 15, FMI was negatively correlated with MVPA (r=-0.235, p<0.001) but not correlated with sedentary behaviour (r=0.042, p=0.284) (lines 200-202). We have also mentioned this correlation in the discussion section.
Include the name of statistical tests and values descriptions in all figures and tables.
We have added the name of statistical tests used in all tables. If by value description, you are referring to value labels, then we are afraid it is not recommended. The figures (1-3) display the estimated values based on trajectories and tables display the actual observed values at each wave. This might vary by a small margin and to avoid any confusions, we believe values in tables are sufficient. Previous papers also follow same recommendations by Nagin et al., 2016; Gallant et al., 2020; Gomes et al., 2017; Hanson et al., 2019.
Overall we are grateful for your valuable suggestions and this has improved the scientific quality of the paper.

Reviewer 3 Report
The present study is the first to jointly investigate the role of moderate-high intensity physical activity (MVPA) and sedentary behaviour (SB) in obesity prevention. The study involved 672 children aged 7, 9, 12 and 15 years. The children were subgrouped according to MVPA and SB, and the propensity of these subgroups to be obese was examined. The final finding of the study is that high MVPA and low SB together protect against obesity, which is consistent with the literature. However, the following suggestions need to be addressed:
- The results mentioned in the abstract are not included in the manuscript: “The third trajectory group (15% of the cohort) that had relatively high MVPA and relatively low SB throughout had lower FMI (-1.7, 95% CI (-3.4 to -1.0) kg/m2, p=0.034) at age 15 compared to the inactive throughout group.” What is the reason for this?
- The Authors are brief and concise in the Introduction of the manuscript. Since the study population is from England, it is worthwhile to refer in a few sentences to the situation of obesity there.
- From the description of the population, it is not clear how this data was collected in relation to the ethnic origin of the participants, the characteristics of their living environment and data on their parents (lifestyle, BMI, age, etc.). Are these data available? If so, have these factors been corrected for in the analysis? If not, it would be worth mentioning the absence of these factors as a limitation.
- One of the requirements of the ANOVA statistical method is a normal distribution of the data. Were the data normally distributed? Was there any data transformation as part of the analysis?
- Table 1 summarises the characteristics of the study population, but the gender distribution is missing. Without this, the table is incomplete.
- Trend analysis is recommended for the statistical analysis of changes over time (waves) in anthropometric and accelerometer data shown in Table 1.
- For figures, the grey dashed lines indicating a 95% CI are difficult to see.
There is a significant difference in the body composition (and therefore FMI) of boys and girls after puberty (and even a little before). This could have a significant impact on the results, so it would be important to include sex ratios at the respective sampling times (waves) and in the subgroups formed by MVPA and SB. Based on the results of the present study, it is not possible to state clearly whether MVPA and SB or sex differences (whether the sex composition of groups 1 and 4 is different) are the reason for the difference in FMI. This phenomenon is a very critical point of the study and needs to be addressed.
Author Response
Reviewer 3:
Open Review
(x) I would not like to sign my review report
( ) I would like to sign my review report
English language and style
( ) Extensive editing of English language and style required
( ) Moderate English changes required
( ) English language and style are fine/minor spell check required
(x) I don't feel qualified to judge about the English language and style
|
Yes |
Can be improved |
Must be improved |
Not applicable |
|
|
Does the introduction provide sufficient background and include all relevant references? |
(x) |
( ) |
( ) |
( ) |
|
Is the research design appropriate? |
(x) |
( ) |
( ) |
( ) |
|
Are the methods adequately described? |
( ) |
(x) |
( ) |
( ) |
|
Are the results clearly presented? |
( ) |
( ) |
(x) |
( ) |
|
Are the conclusions supported by the results? |
( ) |
( ) |
(x) |
( ) |
Comments and Suggestions for Authors
The present study is the first to jointly investigate the role of moderate-high intensity physical activity (MVPA) and sedentary behaviour (SB) in obesity prevention. The study involved 672 children aged 7, 9, 12 and 15 years. The children were subgrouped according to MVPA and SB, and the propensity of these subgroups to be obese was examined. The final finding of the study is that high MVPA and low SB together protect against obesity, which is consistent with the literature. However, the following suggestions need to be addressed:
- The results mentioned in the abstract are not included in the manuscript: “The third trajectory group (15% of the cohort) that had relatively high MVPA and relatively low SB throughout had lower FMI (-1.7, 95% CI (-3.4 to -1.0) kg/m2, p=0.034) at age 15 compared to the inactive throughout group.” What is the reason for this?
Sorry only the overall association was mentioned. But now we have also added the post-hoc pairwise result that was already mentioned in abstract (line 266-267).
- The Authors are brief and concise in the Introduction of the manuscript. Since the study population is from England, it is worthwhile to refer in a few sentences to the situation of obesity there.
Child and adolescent obesity prevalence in England was high and increasing slowly at the time of the study, and was more prevalent in families of low socio-economic status. Prevalence and trends were very similar in the cohort studied here and this has now been summarised at the beginning of the Methods section where the cohort is described more fully (lines (114-118)
- From the description of the population, it is not clear how this data was collected in relation to the ethnic origin of the participants, the characteristics of their living environment and data on their parents (lifestyle, BMI, age, etc.). Are these data available? If so, have these factors been corrected for in the analysis? If not, it would be worth mentioning the absence of these factors as a limitation.
Due to overwhelming number of findings from this study, it was decided to analyse the factors associated with group membership as a separate focused paper. We have added the following statement in the limitation section. Also the conclusion last line states that “future research is warranted to determine predictors that influence group membership.” We are in the process of submitting this paper as well.
- One of the requirements of the ANOVA statistical method is a normal distribution of the data. Were the data normally distributed? Was there any data transformation as part of the analysis?
Yes, we have checked the data for normality before the analysis using data visuals histogram and also Shapiro-Wilk test. Statistical analysis section is now updated.
- Table 1 summarises the characteristics of the study population, but the gender distribution is missing. Without this, the table is incomplete.
Now we have added the gender distribution in Table 1.
- Trend analysis is recommended for the statistical analysis of changes over time (waves) in anthropometric and accelerometer data shown in Table 1.
This is a good suggestion, we have now compared the changes in physical activity and anthropometry data between each age groups. The table 1 is updated.
- For figures, the grey dashed lines indicating a 95% CI are difficult to see.
Thank you for the suggestion. The grey lines are more thicker and visible in high resolution pdf attached.
There is a significant difference in the body composition (and therefore FMI) of boys and girls after puberty (and even a little before). This could have a significant impact on the results, so it would be important to include sex ratios at the respective sampling times (waves) and in the subgroups formed by MVPA and SB. Based on the results of the present study, it is not possible to state clearly whether MVPA and SB or sex differences (whether the sex composition of groups 1 and 4 is different) are the reason for the difference in FMI. This phenomenon is a very critical point of the study and needs to be addressed.
The study of factors or correlates to group membership is a immediate follow up of this study. In fact, we have added this in the recommendation for future research in the conclusion. Gender is one of the factors for sure. Due to overwhelming number of findings from this study, it was decided to analyse the factors associated with group membership as a separate focused paper. We have added the following statement in the limitation section. Also the conclusion last line states that “future research is warranted to determine predictors that influence group membership.” We are in the process of submitting this paper soon.
Overall we are grateful for your valuable suggestions and this has improved the scientific quality of the paper.
Round 2
Reviewer 1 Report
I thank the Authors for their clear explanations and answers to my comments, and I have no additionnal comment.
Author Response
Thank you for the feedback. This has helped us improving the manuscript.
Reviewer 3 Report
I accept the Authors' replies with the exception of the last point. In my opinion, the impact of gender differences in the present study could be very significant (as I have explained in my previous report). This issue cannot be settled by simply referring to the conduct of future research(es). It should definitely be addressed in this manuscript.
- Some analyses by gender may not be possible due to low sample sizes as mentioned in the limitations. However, in the case of Table 1, which shows anthropometry and accelerometer results, this is possible. It would be illustrative to show how the results have changed for boys and girls separately.
- As with Table 1, the gender composition of the subgroups (proportion of boys to girls) should be indicated in Tables 2.
- Furthermore, if no correction is made for gender (in the present study, only for the future one), the possible impact of it on the presented results should be indicated in the limitations.
